# Multi-Task Interactive Robot Fleet Learning with Visual World Models

**Huihan Liu**  **Yu Zhang**  **Vaarij Betala**  **Evan Zhang**

**James Liu**  **Crystal Ding**  **Yuke Zhu**

The University of Texas at Austin

**Abstract:** Recent advancements in large-scale multi-task robot learning offer the potential for deploying robot fleets in household and industrial settings, enabling them to perform diverse tasks across various environments. However, AI-enabled robots often face challenges with generalization and robustness when exposed to real-world variability and uncertainty. We introduce Sirius-Fleet, a multi-task interactive robot fleet learning framework to address these challenges. Sirius-Fleet monitors robot performance during deployment and involves humans to correct the robot's actions when necessary. We employ a visual world model to predict the outcomes of future actions and build anomaly predictors to predict whether they will likely result in anomalies. As the robot autonomy improves, the anomaly predictors automatically adapt their prediction criteria, leading to fewer requests for human intervention and gradually reducing human workload over time. Evaluations on large-scale benchmarks demonstrate Sirius-Fleet's effectiveness in improving multi-task policy performance and monitoring accuracy. We demonstrate Sirius-Fleet's performance in both RoboCasa in simulation and Mutex in the real world, two diverse, large-scale multi-task benchmarks. More information is available on the project website: https://ut-austin-rpl.github.io/sirius-fleet

**Keywords:** Robot Manipulation, Interactive Imitation Learning, Fleet Learning

## 1 Introduction

In recent years, there have been significant advancements in developing robots capable of performing various tasks [1, 2, 3]. The rapid progress in generalist robots holds great potential for deploying robot fleets [4, 5, 6] in households and industrial environments where the robots operate under a generalist multi-task policy. Despite these research advances, robots face challenges with generalization and robustness when deployed in real-world environments, which are often diverse and unstructured. These challenges undermine the safety and reliability of robot systems and limit their applicability in practical scenarios.

To address these challenges, a series of works have been developed on interactive imitation learning (IIL) [7, 8, 9, 10, 11, 12, 13] and interactive fleet learning (IFL) [14, 5, 6]. Prior work on human-in-the-loop learning [7, 13, 15, 10] has proposed involving humans in real-time monitoring and correction to ensure trustworthy deployment. However, these methods often require continuous human supervision. To reduce the high human workload, runtime monitoring approaches [16, 17, 18, 19, 12, 20, 21] have been proposed. These approaches automatically monitor robot performance, identify anomalies, and ask for human control when needed. Methods like out-of-distribution (OOD) detection [20, 21, 18] and failure detection [12] have been introduced to identify anomaly cases of robot execution. However, these methods have primarily been used in single-task settings, limiting their effectiveness for large-scale, multi-task fleet deployment.

We introduce Sirius-Fleet, a multi-task interactive robot fleet learning framework. Sirius-Fleet consists of a multi-task policy and a runtime monitoring mechanism. Sirius-Fleet enables

8th Conference on Robot Learning (CoRL 2024), Munich, Germany.

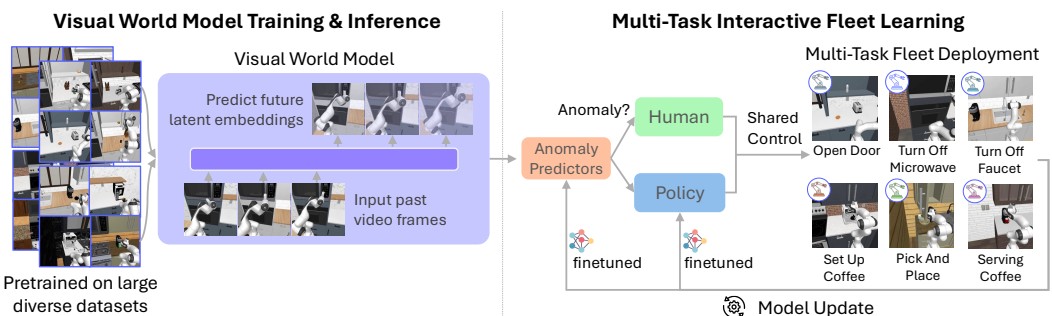

Figure 1: **Overview of SIRIUS-FLEET.** Our framework of multi-task interactive robot fleet learning consists of two stages: 1) Visual World Model Training & Inference, where we pre-train a visual world model on diverse datasets to predict future latent embeddings from past visual observations, and 2) Multi-Task Interactive Fleet Learning, where the pre-trained world model is used to supervise multi-task robot fleet deployment. During deployment, anomaly predictors monitor task performance in real time, soliciting human feedback when necessary. The policy and anomaly predictors are continuously fine-tuned with deployment data, improving task performance over time.

the multi-task policy to be deployed across large robot fleets with the runtime monitoring mechanism, which requests human intervention when anomalies are predicted. SIRIUS-FLEET tackles two challenges. First, SIRIUS-FLEET enables a multi-task policy to be continually updated throughout long-term deployment. Multi-task policy training enables knowledge sharing across tasks and, therefore, improves generalization. Second, SIRIUS-FLEET builds a runtime monitoring mechanism that efficiently supervises multiple tasks simultaneously. To address the first challenge, we train a multi-task policy that can be constantly fine-tuned with deployment data. For the second challenge, we employ a visual world model as the backbone for runtime monitoring, enabling the sharing of learned representations across downstream anomaly predictors for various tasks.

A key challenge in runtime monitoring is to effectively predict future action outcomes, as it allows the system to preempt potential failures before they occur. Recent advancements in world models [22, 23, 24, 25] have shown the capability of simulating future scenarios and predicting future task outcomes. Inspired by this, we develop a visual world model trained on diverse robot trajectories performing a large variety of tasks, which enables the prediction of future task outcomes and helps prevent potential failures. The visual world model is trained by reconstructing image frames from input observations, which allows it to capture fine-grained visual details necessary for precise manipulation. The learned embeddings from the world model are then shared across downstream anomaly prediction tasks. We train two distinct types of anomaly predictors: failure prediction and out-of-distribution (OOD) prediction. The two predictors complement each other — failure prediction predicts failures similar to those identified by humans previously, and OOD prediction captures cases when the robot is in novel, unfamiliar scenarios. The predictors are trained using the frozen embeddings from the visual world model and are continuously fine-tuned during deployment. Unlike prior work that uses fixed prediction thresholds, SIRIUS-FLEET automatically adjusts the anomaly predictors' criteria based on task performance and human feedback. This adaptive threshold aligns with the robot's evolving level of autonomy, resulting in more effective runtime monitoring.

We evaluate our multi-task interactive robot fleet learning framework on large-scale benchmarks in simulation and real-world environments. Our key findings are: 1) our runtime monitoring system effectively supervises diverse multi-task scenarios with $> 95\%$ success rates in overall system performance, 2) the multi-task policy continually improves over time by leveraging deployment data, 3) our anomaly predictors for runtime monitoring outperform baseline methods in accuracy and reduction of human workload. In summary, our contributions are:

1. A framework for multi-task interactive robot fleet learning. The multi-task robot policy efficiently improves over deployment through runtime monitoring and human interaction;

2. A runtime monitoring mechanism based on a visual world model backbone with task-adaptive anomaly prediction thresholds;

3. A demonstration of the high performance of our multi-task fleet learning system in both simulation and real-world environment, achieving on average $> 95\%$ success rates in system performance.

## 2 Related Work

**Multi-Task Robot Learning.** Recent advancements in multi-task robot learning have seen the development of agents capable of performing diverse tasks across various domains [2, 26, 27, 1, 28, 29, 30, 31] These advances have been driven by innovations in policy architectures [1, 32, 33, 34], the availability of large-scale datasets [35, 36, 37, 38], and new robotics benchmarks [32, 39]. Despite this progress, many of these models are deployed as static, one-off implementations, which limits their robustness and generalization in real-world, unstructured environments. In contrast, SIRIUS-FLEET is the first framework for robot manipulation that enables iterative improvement of multi-task policies through human-in-the-loop interaction during deployment.

**Robot Fleet Learning.** The recent large-scale robot deployment and data collection efforts [40, 41, 42] have increased interest in robot fleet learning [43, 4, 5, 44, 45, 6]. This area of research addresses key challenges such as resource allocation [43, 14], decentralized and federated learning [5, 44, 45], and system management [6]. However, two significant challenges remain: supervising large robot fleets with minimal human oversight and enabling multi-task policy improvement over time using deployment data. SIRIUS-FLEET addresses these challenges by combining runtime monitoring with continuous policy updates in a multi-task, interactive fleet learning framework. This ensures that policies improve iteratively based on deployment data and human feedback.

**Interactive Imitation Learning.** Human-in-the-loop methods have been introduced to ensure safe and trustworthy deployment by allowing robots to learn from human interventions during task execution [10, 9, 7, 13, 15, 8, 11]. To reduce the burden of constant human oversight, runtime monitoring techniques have been developed to identify anomalies during task execution [12, 17, 46, 16, 47, 48]. Two main areas of focus are unsupervised out-of-distribution (OOD) detection [18, 49, 19, 50, 20, 21], and failure detection, which can be done either by binary classification [51, 52, 53] or by learning risk functions from trajectory data [12, 14]. Recent advancements in foundation models have also led to the use of Large Language Models (LLMs) and Vision Language Models (VLMs) to identify anomalies for policies based on these models [54, 55, 56]. While prior work has primarily focused on task-specific dynamics models for single-task settings, SIRIUS-FLEET introduces a visual world model trained on diverse datasets before deployment. This model can predict anomalies across various tasks, making SIRIUS-FLEET scalable without additional training during deployment.

## 3 SIRIUS-FLEET: Multi-Task Interactive Robot Fleet Learning

### 3.1 Background

#### 3.1.1 Problem Formulation

We formulate multi-task interactive robot fleet learning as a finite-horizon Markov Decision Process (MDP), where $N$ robots operate in $N$ independent MDPs. The $i$-th robot operates in its respective $i$-th MDP, defined as $\mathcal{M}_i = (S, A, \mathcal{T}, H_i, \mu_i^0, R_i)$, where $S$ is the state space, $A$ is the action space, $\mathcal{T} : S \times A \to S$ is the transition dynamics, $H_i$ is the horizon length, $\mu_i^0$ is the initial state distribution, and $R_i : S \times A \to \mathbb{R}$ is the reward function. In sparse-reward settings, $R_i$ is replaced with a goal predicate $g_i : S \to \{0, 1\}$. The collection of MDPs, $\{\mathcal{M}_i\}_{i=1}^N$, can be reformulated as a single unified MDP with shared state space $S$, action space $A$, and transition function $\mathcal{T}$. Data from all robots are aggregated to learn a unified multi-task policy $\pi(a \mid s, g_i)$, which maximizes the expected return: $\max_\pi J(\pi) = \mathbb{E}_{s_t, a_t \sim \pi, \mu_0} \left[ \sum_{t=1}^H g_i(s_t) \right]$.

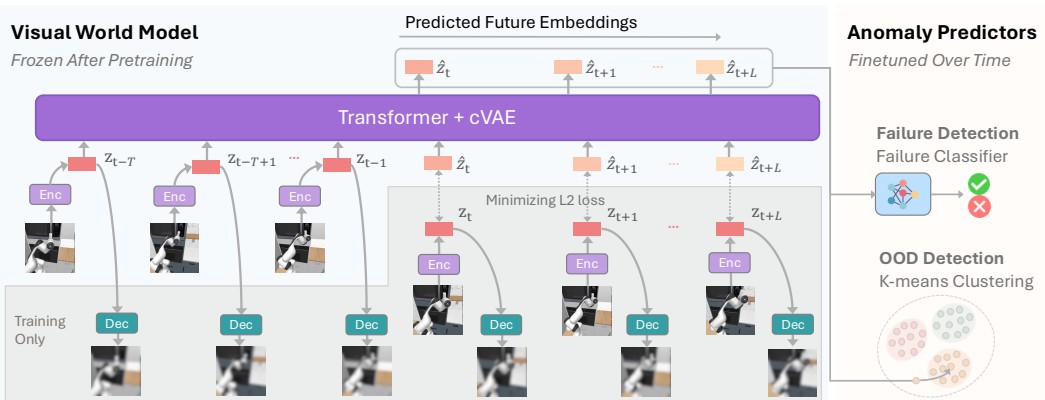

Figure 2: **Model Architecture.** The visual world model comprises a UNet-based encoder and decoder combined with a cVAE- and Transformer-based prediction model. This architecture allows the world model to predict future embeddings from the current state. The learned representations are then used for anomaly predictions, including failure and OOD prediction.

### 3.1.2 Multi-Task Interactive Fleet Deployment

We consider an interactive learning framework [7, 13, 17] for a fleet of robots [4], where learning and deployment happens iteratively with a human policy $\pi_H$ in the loop. Each robot bootstraps its initial policy $\pi_0$ via behavioral cloning from a set of human demonstrations, $\mathcal{D}^0$. Starting from deployment round $i = 1$, each robot executes tasks using policy $\pi_i$ with runtime monitoring, supervised by anomaly predictors $E_i$. During policy execution, $E_i$ determines whether the current state may lead to anomalies. Upon predicting a potential anomaly, the system signals the human to monitor the process. While monitoring, the human can choose to actively intervene [57] and take control if necessary, allowing the human policy $\pi_H(s, g)$ to override the robot policy $\pi_i(s, g)$. The resulting trajectories $\tau = (s_t, a_t, r_t, c_t)$ are added to the data buffer $\mathcal{D}'$ of the current deployment round, where $c_t$ indicates whether timestep $t$ is controlled by human action. The data buffer is then updated as $\mathcal{D}^{i+1} = \mathcal{D}^i \cup \mathcal{D}'$, which is used to train the next-round policy $\pi_{i+1}$ and anomaly predictors $E_{i+1}$.

### 3.2 Runtime Monitoring for Multi-Task Fleet Deployment

We present SIRIUS-FLEET's runtime monitoring mechanism, which supervises multiple tasks simultaneously across diverse environments during deployment. The system is designed to meet three key goals: 1) generalization—the anomaly predictors are built on shared embeddings from the visual world model, allowing them to be used across different tasks; 2) task adaptability—it adjusts dynamically to the evolving progress of each task during deployment; and 3) failure preemption—it predicts anomalies before they occur, enabling timely prediction and intervention. To achieve this, we train a visual world model that simulates task progress and supports anomaly prediction across various tasks, as illustrated in Figure 2.

### 3.2.1 Training the Visual World Model

Inspired by recent advances in world models [22, 23, 24, 25], we train a visual world model on diverse robot trajectory frames to predict future task outcomes and prevent potential failures. The visual world model, trained by reconstructing image observations, predicts future latent embeddings. This world modeling approach is effective for several reasons: 1) pixel reconstruction is a readily available form of supervision, allowing the model to learn without manual annotations; 2) reconstructing image frames helps the model capture the fine-grained visual details necessary for precise manipulation tasks, and 3) it helps to develop the model's ability to predict changes in the robot's visual environment over time.

SIRIUS-FLEET trains an autoregressive visual world model $\mathcal{W}$ as the backbone for downstream anomaly prediction. The world model $\mathcal{W} = (E_\gamma, D_\lambda, T_\psi)$ consists of an encoder $E_\gamma$, a decoder

$D_\lambda$, and a conditional next state prediction model $T_\psi$. In training, $E_\gamma$ encodes image observation $x_t$ at timestep $t$ into latent embedding $z_t$. The decoder $D_\lambda$ reconstructs $z_t$ to image $\hat{x}_t$ and minimizes the image reconstruction L2 loss. $T_\psi$ inputs the history of $T$ timesteps of embeddings $(z_{t-T}, z_{t-T+1}, \ldots z_t)$, and outputs $\hat{z_{t+1}}$. It autoregressively predicts $(\hat{z_{t+1}}, \hat{z_{t+2}}, \ldots \hat{z_{t+L}})$ for $L$ steps into the future using the last $T$ timesteps of embeddings and reconstructed embeddings, and minimizes the embedding reconstruction loss between $(\hat{z_{t+1}}, \hat{z_{t+2}}, \ldots \hat{z_{t+L}})$ and $(z_{t+1}, z_{t+2}, \ldots z_{t+L})$. $E_\gamma$ and $D_\lambda$ are implemented with UNet [58, 33], and $T_\psi$ uses conditional Variational Autoencoder (cVAE) [59]. $T_\psi$ is jointly trained with $E_\gamma, D_\lambda$ on the same latent space.

We use a stochastic latent space rather than a deterministic one since the stochastic latent space of cVAE supports multiple future sampling and facilitates better prediction [60, 17]. We use transformer architecture [61] for the encoder, decoder, and prior network for the cVAE in $T_\psi$.

### 3.2.2 Building the Downstream Anomaly Predictors

The visual world model captures changes in task outcomes over time. Its learned representation can be used for the anomaly predictors to predict future anomalies. Since individual tasks vary, we train task-specific anomaly predictors on the frozen representation to predict failures and OOD anomalies.

**Failure Prediction.** We train a failure classifier $F_\sigma$ for each task using the frozen image embeddings from the visual world model. Failure labels are from trajectories $\tau = (s_t, a_t, r_t, c_t)$ in the last round $\mathcal{D}'$ (see Section 3.1.2), where $c_t$ marks human interventions (human). The trajectory segment before each intervention are labeled as failures (failure) [13, 17]. The classifier is trained using a cross-entropy loss $\mathcal{L}_F = -\sum_{i=1}^{n} y_i \log(\hat{y}_i)$ with balanced sampling, where $y_i \in \{\text{rollout}, \text{failure}, \text{human}\}$. The failure classifier is a small, computationally efficient model trained on frozen world model embeddings (training time 1.5 hour).

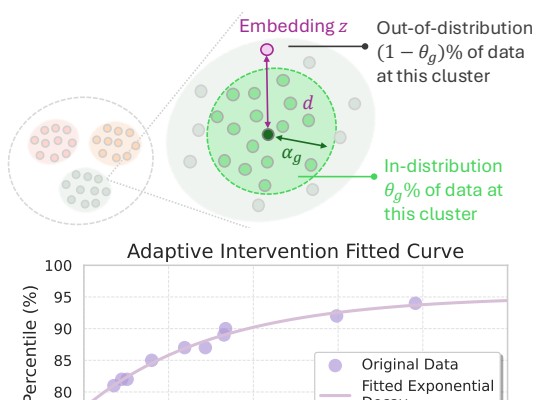

**Out-of-Distribution (OOD) Prediction.** We identify OOD states using k-means clustering. The frozen visual world model $\mathcal{W}$ generates embeddings from sampled trajectories in the data buffer. We use Principal Component Analysis (PCA) to reduce the embedding dimensions to $l$ and calculate $c$ k-means centroids for each task. To predict OOD for an embedding $z$, we reduce its dimensions, find the nearest centroid, and calculate its L2 distance $d$. A state is

Figure 3: **Adaptive Decision Boundaries.** Top: OOD Prediction Boundary. The threshold $\theta_g$, determined by the human intervention ratio, sets the distance threshold $\alpha_g$. A sample is identified as OOD if its embedding's distance $d$ from the cluster centroid exceeds $\alpha_g$. Bottom: Fitting function for optimal $\theta_g$ based on the human intervention ratio $p_H$. The x-axis shows $1 - p_H$, representing the autonomous rollout ratio.

identified as OOD if $d$ exceeds the task threshold $\alpha_g$. $\alpha_g$ is determined by $d_g^\theta$, which is the distance of the top $\theta_g$ percentile of the distances to the nearest centroids from the validation latent embeddings. For efficiency, we perform task-specific k-means clustering on image embeddings for the specific task without training additional models.

**Adaptive Decision Boundaries.** Anomaly predictors should adjust their anomaly prediction threshold as robot performance improves during deployment. In a multi-task setting, the varying task performances across different tasks pose additional challenges. We adjust anomaly predictors by loosening the decision boundaries for high-performing tasks and tightening them for low-performing tasks. For failure prediction, we finetune it using the most recent round of deployment data $\mathcal{D}'$, with human intervention labels reflecting the updated human perceived risk.

The decision boundary expands for OOD prediction as the robot encounters more in-distribution states. We sample from $D^i$, the aggregated deployment data for all rounds, for k-means clustering. For each new embedding $z$, we use a threshold $\alpha_g$ for its distance $d$ to its nearest centroid, which depends on $\theta_g$. We update the prediction threshold $\alpha_g$ based on the human intervention ratio $p_H$, which acts as a proxy for policy performance. We fit the exponential decay function, $\theta_g = a + be^{cp_H}$, and obtain the parameters by fitting the curve to a set of calibration trajectories. Empirically, $a = 95.2$, $b = -17.7$ and $c = -3.2$. We found this function to be robust across all rounds and all tasks from simulation to real-world experiments, and we used the same function and set of hyperparameters throughout the simulation and real-world experiments.

At deployment, the embedding $z$ is computed from observations, and the distance $d$ from the nearest centroid is compared to $\alpha_g$. If $d > \alpha_g$, the sample is identified as OOD. Figure 3 (Top) shows the decision boundary, while Figure 3 (Bottom) shows the fitted curve for $\theta_g$.

### 3.2.3 SIRIUS-FLEET in Operation

**Continual Model Improvement.** While the visual world model $\mathcal{W}$ is trained once and frozen, the policy and the anomaly predictors are continually finetuned over deployment: the policy is updated with the data from previous rounds using weighted sampling, the failure classifier for failure prediction is finetuned with the most recent intervention labels, and the K-means clustering in OOD prediction expands its latent space coverage after each round.

Figure 4: **Policy Architecture.** The multi-task policy is a Transformer that processes images, proprioceptive data, and task language embeddings. It uses a Gaussian Mixture Model (GMM) to output robot actions.

**Anomaly Predictors at Runtime.** During deployment, the visual world model predicts each task's future embeddings over $L$ steps. We generate $N$ possible future scenarios by sampling from the cVAE latent space and making $L$-step predictions $N$ times. Each predicted future embedding is then assessed individually by the anomaly predictors. For failure prediction, we compute the average failure score across future embeddings. For OOD prediction, we calculate the average distance of each future state to its nearest cluster centroid and compare the final average distance against the OOD threshold.

**Multi-Task Policy Training.** We train a Transformer-based [61] multi-task policy, as shown in Figure 4. The policy inputs image observations, robot proprioceptive data, language task goals, and output robot actions. The design follows that in RoboMimic [32] and RoboCasa [39].

## 4 Experiments

In our experiments, we aim to address the following questions: 1) How well does SIRIUS-FLEET continue to improve its policy during deployment with effective runtime monitoring? 2) How does SIRIUS-FLEET's runtime monitoring performance compare with baseline methods? 3) Are the anomaly prediction made by SIRIUS-FLEET at the right moments, qualitatively?

### 4.1 Evaluation Setup

To measure the effectiveness of SIRIUS-FLEET's multi-task fleet learning, we evaluate how both the policy and the runtime monitoring perform and evolve over time. Following prior works [62, 13, 17], we evaluate the system in a human-in-the-loop setting through rounds of iterative deployment.

**Evaluation Setting.** Our evaluation spans three rounds of policy updates and runtime monitoring. Full human supervision is required in the first round since no anomaly predictors have been trained

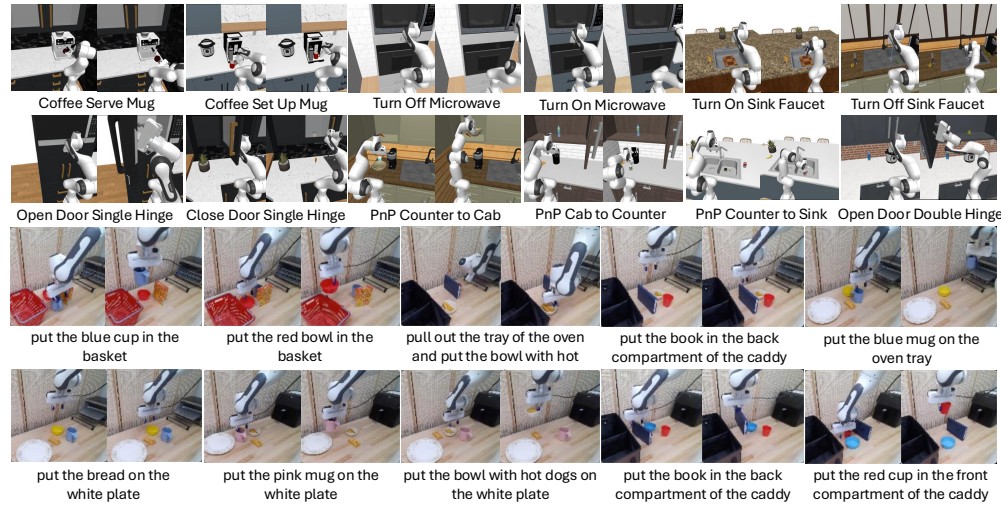

Figure 5: **RoboCasa Simulation Tasks and Mutex Real-World Tasks.** We evaluate policy learning and runtime monitoring using 12 tasks from the RoboCasa benchmark in simulation and 10 tasks from the Mutex benchmark in real-world environments.

yet. Human supervision is only requested in the next two rounds when an anomaly is predicted. After each round, the collected deployment data and intervention labels are used to train the policy and anomaly predictors for the next round. The initial policy is trained with 50 human demonstrations per task over 1200 epochs. We collect 100 rollouts per task for each round to ensure consistent data size and finetune the policy for 400 epochs.

**Evaluation Metrics.** We evaluate SIRIUS-FLEET's policy using **Autonomous Performance**, which measures the policy's ability to achieve its goals without human intervention. After each round, the policy is finetuned on the newly collected data, and we evaluate its success rate without runtime monitoring to assess performance improvements over time. To evaluate runtime monitoring, we use **Combined Policy Performance (CPP)** and **Return of Human Effort (ROHE)**, following [12, 17]. CPP measures the overall system success rate under monitoring, reflecting the effectiveness of human-robot collaboration in preventing anomalies. ROHE assesses the efficiency of human intervention by comparing the policy's success rate against the amount of human intervention effort, calculated as the ratio of policy performance to the number of interventions: Normalized ROHE = $\frac{\mathbb{E}_\tau [\sum_{t=0}^{T_\tau} r_t^\tau]}{1 + \frac{H}{T}}$. The goal is to maximize performance while minimizing human workload.

**Evaluation Environments.** For simulation, we use RoboCasa [39], a visually diverse benchmark with various objects, layouts, and scenes. RoboCasa contains both a large dataset using MimicGen [38], an automated trajectory generation method (5k trajectories per task), and a human demonstration dataset (50 human demonstrations per task). The visual world model is trained on 20 task suites of MimicGen data, and the policy is trained on 12 task suites of human demonstration data. We use Mutex [27] for real-world experiments, a commonly used benchmark for multi-task learning. The visual model is trained on 50 tasks, and the policy is trained on 10 tasks across 5 task suites. We use an OSC controller with 7D action space (x-y-z position, yaw-pitch-roll orientation, and gripper).

## 4.2 Evaluation Results

**System Performance Over Time.** Figure 6 presents the performance of SIRIUS-FLEET over several deployment rounds. The autonomous policy improves consistently, showing a 13% increase in simulation and 45% in real-world settings. SIRIUS-FLEET maintains high combined policy performance, exceeding 95% in both environments, demonstrating the effectiveness of runtime monitoring and timely human intervention. Additionally, SIRIUS-FLEET shows improved ROHE performance

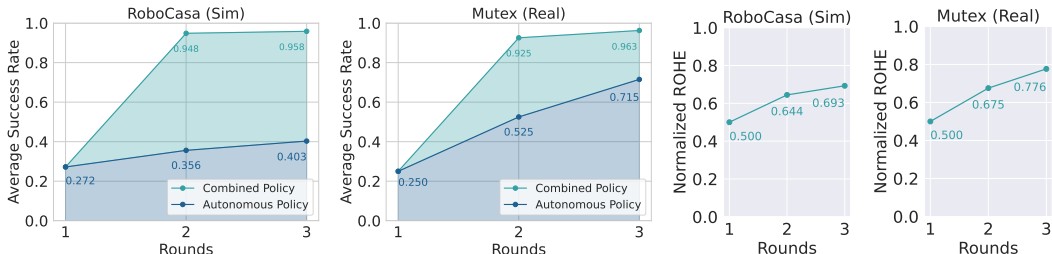

Figure 6: **System Performance Results.** SIRIUS-FLEET shows consistent improvement throughout deployment for Combined and Autonomous Policy Performance and for Return of Human Effort (ROHE). The runtime monitoring ensures high combined policy performance and ROHE over time.

over time. The continuous fine-tuning of anomaly predictors and adaptive thresholds leads to fewer predicted anomalies and reduced human intervention.

**Baseline Comparison.** We compare SIRIUS-FLEET's multi-task runtime monitoring against three baseline methods: **MoMart** [20], **PATO** [21], and **ThriftyDAgger** [12]. MoMart uses VAE reconstruction loss for OOD detection, PATO combines ensemble policy variance with VAE reconstruction for predicting future image goals, and ThriftyDAgger merges OOD detection and failure detection using risky Q-function values.

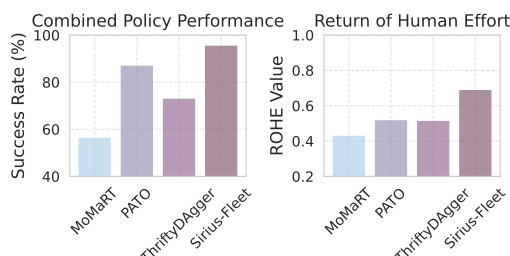

Figure 7: **Baseline Comparison.** SIRIUS-FLEET surpassed the baselines in both combined policy performance and ROHE.

Since these baselines are designed for single-task environments, we evaluated them on five tasks from different categories in RoboCasa, training separate models for each task. All baselines used fixed detection thresholds, while ThriftyDAgger adjusts thresholds based on a target intervention ratio. For fairness, we applied the same intervention ratio across all comparisons. As shown in Figure 7, SIRIUS-FLEET consistently outperforms the baselines in policy performance and ROHE. The fixed thresholds used by the baselines were ineffective in handling varying task distributions, resulting in excessive interventions or decreased performance across tasks.

**Ablations.** We conducted ablation studies to explore the following key questions: 1) How important is the design of the multi-task visual world model, as opposed to the single-task design? 2) What are the individual impacts of the Failure Prediction and OOD Prediction components? 3) How critical is the design of the multi-task policy? The results of these studies are detailed in Appendix 6.2.

## 5   Conclusion

We introduce SIRIUS-FLEET, a framework for multi-task interactive robot fleet learning that combines a multi-task policy with runtime monitoring. Driven by a visual world model and task-adaptive anomaly predictors, SIRIUS-FLEET improves policy performance and reduces human intervention through timely anomaly prediction.

**Limitations.** SIRIUS-FLEET is best suited for quasi-static manipulation tasks, where anomalies can be easily corrected with teleoperation. However, applying SIRIUS-FLEET to dynamic tasks could be challenging. Also, our experiments involve a small group of five human operators; future work could expand this by conducting large-scale human studies to understand the effects of diverse human interventions on runtime monitoring and policy training. Lastly, our evaluations used a single embodiment (Franka Emika Panda arm). Future research can extend SIRIUS-FLEET to cross-embodiment fleet learning, making it more generalizable across different robot platforms [36].

**Acknowledgments**

We thank Zhenyu Jiang, Rutav Shah, Yifeng Zhu, and Shuijing Liu for providing helpful feedback on this manuscript. We also thank Xixi Hu, Zhenyu Jiang, Bo Liu, Yue Zhao, and Lizhang Chen for their fruitful discussions. We thank Soroush Nasiriany, Abhishek Joshi, and the Robo-Casa team for their support in the simulation environments and rendering. This work was partially supported by the National Science Foundation (FRR2145283, EFRI-2318065), the Office of Naval Research (N00014-22-1-2204, N00014-24-1-2550), and the DARPA TIAMAT program (HR0011-24-9-0428).

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

# 6 Appendix

## 6.1 Table of Contents

## 6.2 Ablations

*Multi-Task World Model vs. Single-Task World Model.* We empirically show how the multi-task world model is more generalizable than the single-task version in terms of future latent state prediction accuracy. Accurate future latent state prediction is essential for effective error prediction, as the downstream error predictors directly use the predicted future states. We compared the performance of SIRIUS-FLEET's multi-task world model against six single-task world models in terms of mean squared error (MSE) for future latent state prediction. As shown in Table 1, the multi-task world model consistently outperforms the single-task models across all tasks. This indicates that the multi-task world model is crucial for providing more accurate state predictions.

*Ablation of Failure and OOD Prediction.* We examine the impact of OOD and failure prediction components on human intervention overlap accuracy: We have a human operator fully supervise the robot execution of a policy and can intervene whenever an unsafe state is observed. We then apply the learned error predictors to the collected trajectories and compare failure states identified by the error predictors with those intervened by the human operator. This shows how the error predicted aligns with humans' risk assessment.

As shown in Table 3, SIRIUS-FLEET's combined use of OOD and failure prediction significantly outperforms using either component alone across all tasks. This suggests that both accurate OOD prediction and failure prediction are critical to the overall performance of the policy, as they complement each other in identifying potential errors during deployment.

*Multi-task Policy vs. Single-task Policy.* We show that training on multi-task settings improves the overall performance of robot policy than training on individual single tasks. We show the policy success rate (%) for 7 different tasks in Figure 8, where we compare the multi-task policy trained on all tasks with single-task policies trained on individual tasks. For most tasks, SIRIUS-FLEET's multi-task policy performs better than the single-task policy trained on that specific task. This aligns with the observation in prior works [31, 36, 2] that multi-task training helps the policy generalize better.

| Task | Single Task | Multi Task |
|------|-------------|------------|
| CloseDoorSingleHinge | $3.3 \times 10^{-4}$ | $1.5 \times 10^{-4}$ |
| PnPCabToCounter | $3.1 \times 10^{-4}$ | $1.7 \times 10^{-4}$ |
| TurnOnMicrowave | $1.7 \times 10^{-4}$ | $1.3 \times 10^{-4}$ |
| TurnOnSinkFaucet | $3.2 \times 10^{-4}$ | $1.2 \times 10^{-4}$ |
| CoffeeSetupMug | $4.4 \times 10^{-4}$ | $1.8 \times 10^{-4}$ |

Table 1: **Multi-task World Model vs. Single-task World Model Comparison.** We show the MSE loss ($\downarrow$) for future latent states prediction, where SIRIUS-FLEET 's multi-task world model is compared with 6 different world models trained on 6 single tasks. The multi-task world model consistently gives more accurate future latent prediction than the world models trained with single-task data.

| Task | SIRIUS-FLEET (OOD) | SIRIUS-FLEET (Failure) | **SIRIUS-FLEET** |
|---|---|---|---|
| CoffeeSetupMug | 85.1 | 87.0 | **99.4** |
| PnPCounterToCab | 43.1 | 56.2 | **73.7** |
| TurnOffMicrowave | 54.9 | 50.0 | **74.5** |
| TurnOffSinkFaucet | 11.3 | 38.8 | **45.0** |
| OpenDoorDoubleHinge | 42.8 | 63.3 | **78.4** |

Table 2: **Human Intervention Overlap Accuracy (%) for Different Tasks.** SIRIUS-FLEET (using OOD + Failure combined) achieves better performance than using OOD prediction only or Failure prediction only.

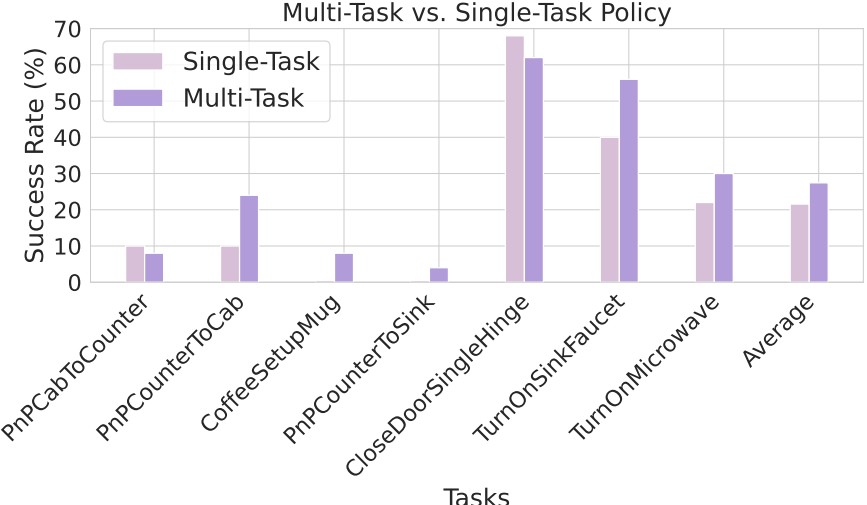

Figure 8: **Multi-task Policy vs. Single-task Policy Comparison.** We show the policy success rate (%) for different tasks, comparing the multi-task policy trained on all tasks with single-task policies trained on individual tasks. SIRIUS-FLEET 's multi-task policy gives overall better performance. Note that for CoffeeSetupMug and PnPCounterToSink, the single-task policy gives 0% success rate.

Table 3: Hyperparameters: World Model

| Group | hyperparameter | Value |
|---|---|---|
| **UNet Encoder** | image output activation | Sigmoid |
| | image observations | [workspace, wrist] |
| | observation fusion method | concat |
| | input channels | 3 |
| | output channels | 3 |
| | latent channels | 4 |
| | block output channels | [32, 64] |
| | layers per block | 1 |
| | activation function | SiLU |
| | normalization number of groups | 32 |
| **Dynamics Model** | history length | 10 |
| | number of futures sampled | 20 |
| **GMM Prior** | latent dimension | 16 |
| | learnable | True |
| | number of GMM nodes | 10 |
| **Transformer Architecture for VAE Encoder, Decoder, Prior** | context length | 20 |
| | embed dimension | 512 |
| | number of layers | 6 |
| | number of heads | 8 |
| | embedding dropout | 0.1 |
| | attention dropout | 0.1 |
| | block output dropout | 0.1 |
| | sinusoidal embedding | False |
| | activation | GeLU |
| | causal | False |

## 6.3 Implementation Details

We present the details of hyperparameters for training the world model, policy, and error predictors in the following table:

- World Model Architecture: Table 3;
- Policy Architecture: Table 4;
- Training: Table 5;
- Failure Prediction: Table 6;
- OOD Prediction: Table 7.

Table 4: Hyperparameters: Policy

| Group | Hyperparameter | Value |
|---|---|---|
| **Transformer Policy** | context length | 20 |
| | embed dimension | 512 |
| | mumber of layers | 6 |
| | number of heads | 8 |
| | embedding dropout | 0.1 |
| | attention dropout | 0.1 |
| | block output dropout | 0.1 |
| | sinusoidal embedding | False |
| | activation | GeLU |
| | causal | False |
| **GMM Head** | number of modes | 5 |
| | min std | 0.005 |
| | std activation | Softplus |
| | low noise eval | True |
| **Image Encoder** | feature dimension | 64 |
| | backbone class | ResNet18ConvFiLM |
| | backbone pretrained | False |
| | backbone input coord conv | False |
| | pool class | SpatialSoftmax |
| | pool number of keypoints | 32 |
| | pool learnable temperature | False |
| | pool temperature | 1.0 |
| | pool noise std | 0.0 |
| **Image Augmentation** | class | CropRandomizer |
| | crop height | 116 |
| | crop width | 116 |
| | number of crops | 1 |
| | positional encoding | False |

Table 5: Hyperparameters: Training

| Group | Hyperparameter | Value |
|---|---|---|
| **World Model** | optimizer type | Adam |
| | initial learning rate | 0.0001 |
| | learning rate decay factor | 0.1 |
| | epoch schedule | [] |
| | scheduler type | Constant |
| **Policy** | optimizer type | AdamW |
| | initial learning rate | 0.0001 |
| | learning rate decay factor | 1.0 |
| | epoch schedule | [100] |
| | scheduler type | Constant with warmup |

Table 6: Hyperparameters: Failure Prediction

| Group | Hyperparameter | Value |
|---|---|---|
| **Transformer Architecture** | context length | 10 |
| | embed dimension | 512 |
| | number of layers | 6 |
| | number of heads | 8 |
| | embedding dropout | 0.1 |
| | attention dropout | 0.1 |
| | block output dropout | 0.1 |
| | sinusoidal embedding | False |
| | activation | GeLU |
| | causal | False |
| **Failure Predictor** | predict on future | True |
| | threshold count | 2 |
| | evaluation index | [7,8,9] |
| | use probability | False |

Table 7: Hyperparameters: OOD Prediction

| Group | Hyperparameter | Value |
|---|---|---|
| **OOD Predictor** | number of future steps | 20 |
| | predict on future | True |
| | distance metric | k-means |
| | train k-means | False |
| | percentile | see Section 3.2.2 |

## 6.4   Qualitative Analysis and Discussion

Can SIRIUS-FLEET catch important errors and give meaningful intervention timings? We conduct a qualitative analysis of where the error predictors predict error during robot execution using *active human monitoring*. Specifically, we have a human operator who fully supervises the robot's policy execution and can intervene whenever an unsafe state is observed. We then apply the learned error predictors to the collected trajectories and compare failure states identified by the error predictors with those intervened by the human operator. This shows how the error predicted aligns with humans' risk assessment.

We show examples of the human intervention region and errors predicted by SIRIUS-FLEET and the baselines in Figure 9 to 14. We use the same trajectory for all methods. The three instances humans gave interventions are: 1) the robot is not aiming at the object it is grasping; 2) the robot pauses at grasping; 3) the arm is too stiff and does not bend down. We visualize if the errors given by SIRIUS-FLEET and the baselines are aligned with the intervention humans gave.

The green area indicates times of actual human intervention; the blue plot indicates the error value predicted by the different methods. We also show the video frame at the corresponding timings to illustrate the failure modes captured (or missed) by the different methods.

Note that SIRIUS-FLEET, PATO, and ThriftyDAgger each have two separate components, so we visualize each individually. PATO and ThriftyDAgger share the same ensemble uncertainty component, so we visualize one of them.

As shown in Figure 9 and 10, our method can capture a similar error pattern to that the human judges. MoMaRT (Figure 11) is prone to predicting false positives due to pixel changes. PATO (VAE part, Figure 12) often predicts false negatives; PATO (ensemble part, Figure 13) often has high oscillations and extreme values. ThriftyDAgger's (Figure 14) risk Q function is data-intensive to train and often has generalization errors; it cannot accurately reflect the task progress and local failure modes.

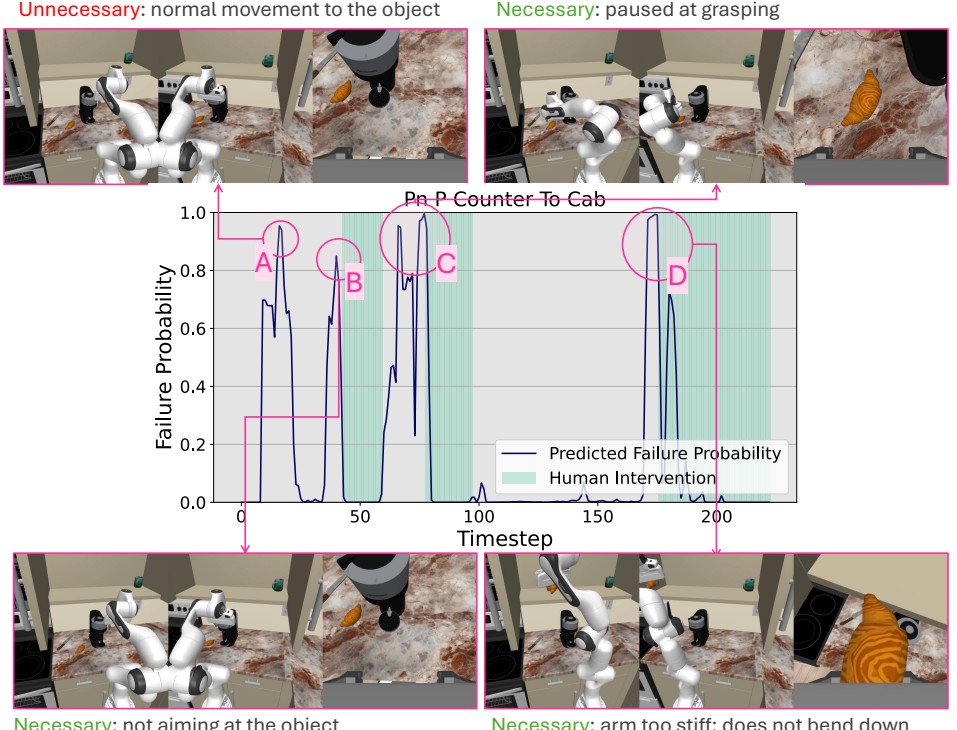

Figure 9: **SIRIUS-FLEET: Failure Only.** The y-axis shows the predicted failure probability from the failure classifier, where a higher value indicates a greater likelihood of failure.

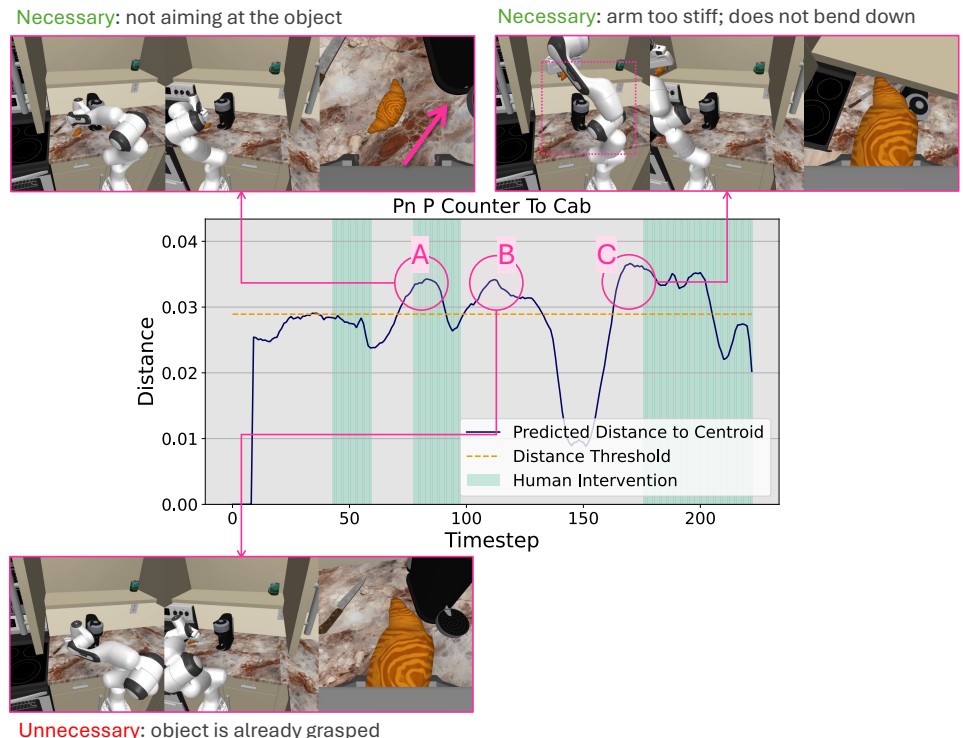

Figure 10: **SIRIUS-FLEET: OOD Only.** The y-axis represents the predicted distance to the centroid. It is predicted as OOD if this distance exceeds a threshold (indicated by the yellow dotted line).

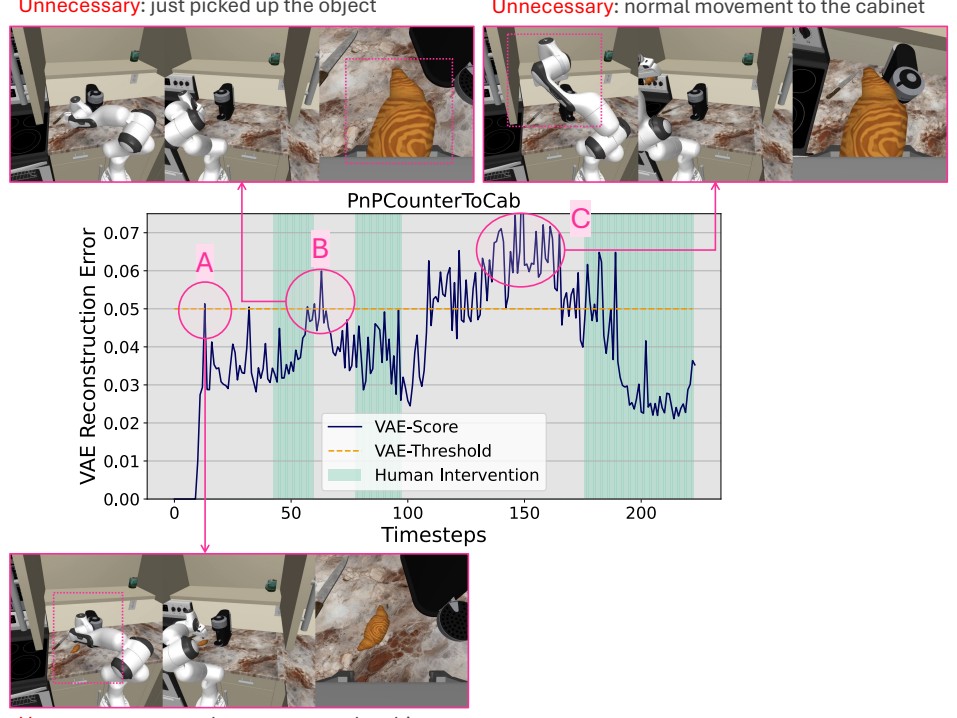

Figure 11: **MoMaRT.** The y-axis represents the VAE reconstruction error (MSE loss) of the current image observation. It is predicted as an error if this loss exceeds a threshold (indicated by the yellow dotted line).

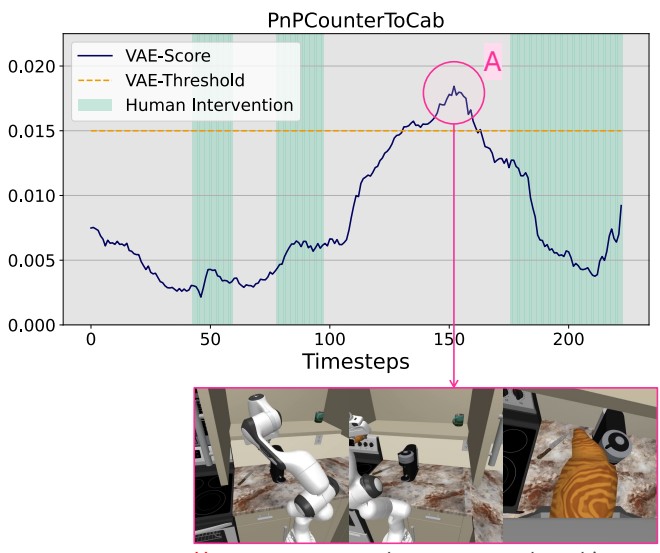

Figure 12: **PATO: VAE Only.** The y-axis represents the variance of the VAE reconstruction of the future image. It is predicted as an error if this value exceeds a threshold (indicated by the yellow dotted line).

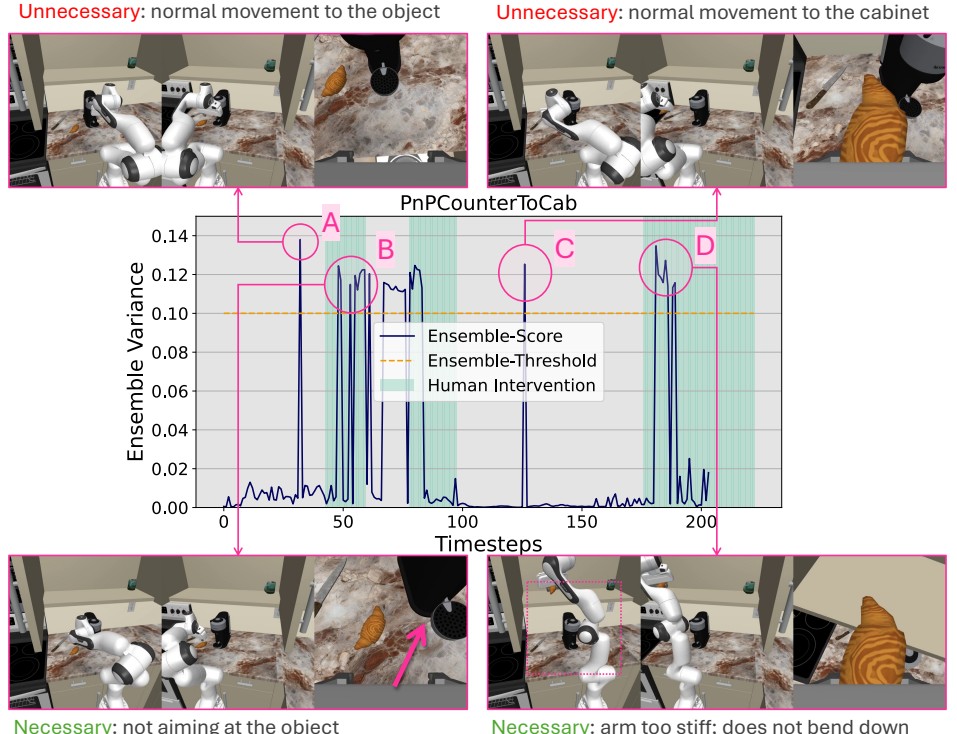

Figure 13: **PATO: Ensemble Only.** The y-axis represents the variance of the ensemble policies. It is predicted as an error if this value exceeds a threshold (indicated by the yellow dotted line).

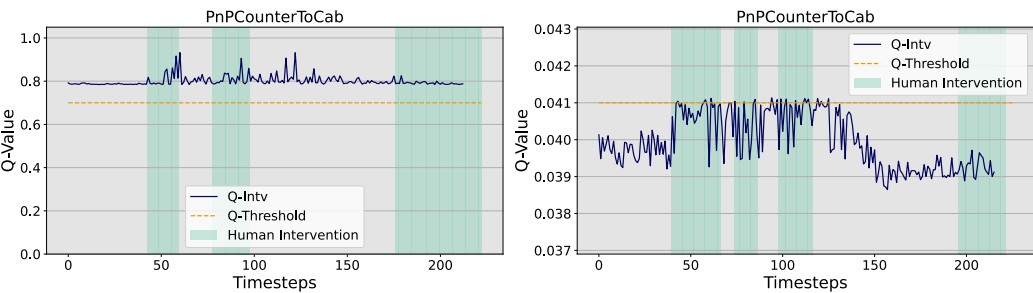

Figure 14: **ThriftyDAgger: Risk Q-Function.** The y-axis represents the q-value of the current state-action pair, where a higher q-value means less risk of failure. The video frame visualization is omitted here since no clear error modes have been predicted.

## 6.5 Additional Details on Tasks

### 6.5.1 Real robot experiments

**Dataset.** We use the Mutex [27] dataset for real robot experiments. The Mutex dataset is a diverse multi-task dataset containing 50 different tasks from 8 task suites (where one task suite is a distinct set of objects and receptacles), specified with natural language. More details on the Mutex dataset can be found in the original Mutex [27] paper. We use the original 50 tasks for training the visual world model; for human workload consideration, we sample 10 of the tasks (among 5 task suites) for multi-task policy learning and runtime monitoring at deployment. The list of 10 tasks for policy learning and runtime monitoring experiments is shown in Figure 15.

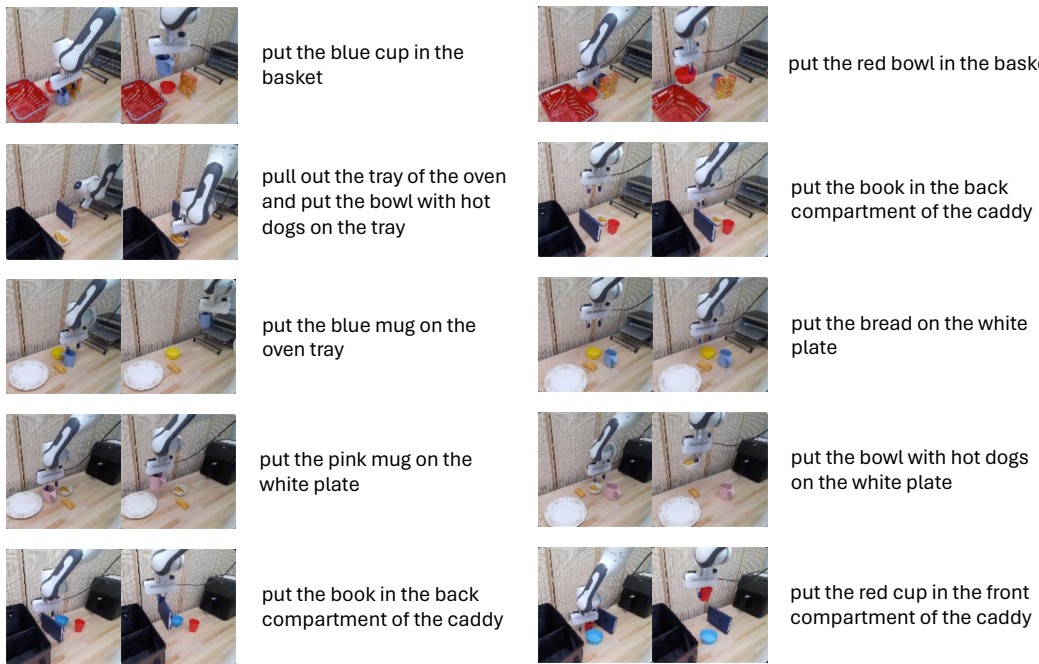

Figure 15: **Mutex Real World Tasks.** We use 10 tasks covering 5 task suites in the Mutex dataset for policy learning and runtime monitoring.

**Training.** To train the visual world model, we use 30 demonstrations for each of the 50 tasks. To train the multi-task policy, 30 demonstrations per task are used to bootstrap the initial BC policy; then, for each of the human-in-the-loop deployment round for runtime monitoring (see Evaluation Setting in Section 4.1), 60 robot rollouts per task are collected for each round.

For the initial BC policy, we train the BC transformer policy for 2000 epochs. For each subsequent round, we finetune the initial BC policy on the newly aggregated data for another 800 epochs. We use this finetuned BC policy checkpoint to perform runtime monitoring experiments.

**Evaluation.** We use 1 seed and the checkpoint at a fixed epoch (epoch 2000 for the initial policy, epoch 800 for the finetuned per-round policy) to evaluate the real robot policy. We evaluate all 10 tasks, conduct 20 trials per task for 200 trials, and report the average success rate across tasks. We report the per-task autonomous policy performance in Figure 17 and per-task combined policy performance in Figure 18 for reference.

### 6.5.2 Simulation

**Dataset.** We use RoboCasa [39] as our simulation environments, which contains a diverse range of tasks *We note one important difference in the environment definition in RoboCasa: the environments are defined by a group of tasks that has similar task semantics, rather than by one single task of a*

*fixed set of scenes and objects.* For example, The `CoffeeServeMug` environment can contain a combination of diverse backgrounds, layouts, coffee machine types, and coffee mug types, making it a challenging environment for generalization. As noted in Section 4.1, We use all 20 suites to train the visual world model and 12 of them for policy learning and runtime monitoring. The list of 10 tasks for policy learning and runtime monitoring experiments is shown in Figure 16.

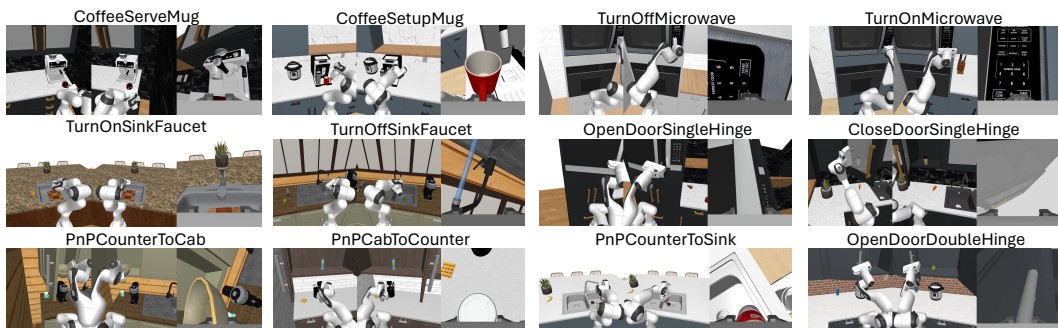

Figure 16: **RoboCasa Simulation Tasks.** We use 12 tasks in the RoboCasa benchmark for policy learning and runtime monitoring.

**Training.** To train the visual world model, we use 50 demonstrations for each of the 20 tasks suites. To train the multi-task policy, 50 demonstrations per task suite are used to bootstrap the initial BC policy; then, for each of the human-in-the-loop deployment round for runtime monitoring (see Evaluation Setting in Section 4.1), 100 robot rollouts per task are collected for each round.

For the initial BC policy, we train the BC transformer policy for 1000 epochs. For each subsequent round, we finetune the initial BC policy on the newly aggregated data for another 400 epochs. We use this finetuned BC policy checkpoint to perform runtime monitoring experiments.

**Evaluation.** We use 2 seeds and checkpoints at a fixed epoch (epoch 1000 for the initial policy, epoch 400 for the finetuned per-round policy) to evaluate the simulation policy. We evaluate all 12 tasks, conduct 50 trials per task for a total of 600 trials, and report the average success rate across tasks. We report the per-task autonomous policy performance in Figure 19 and per-task combined policy performance in Figure 20 for reference.

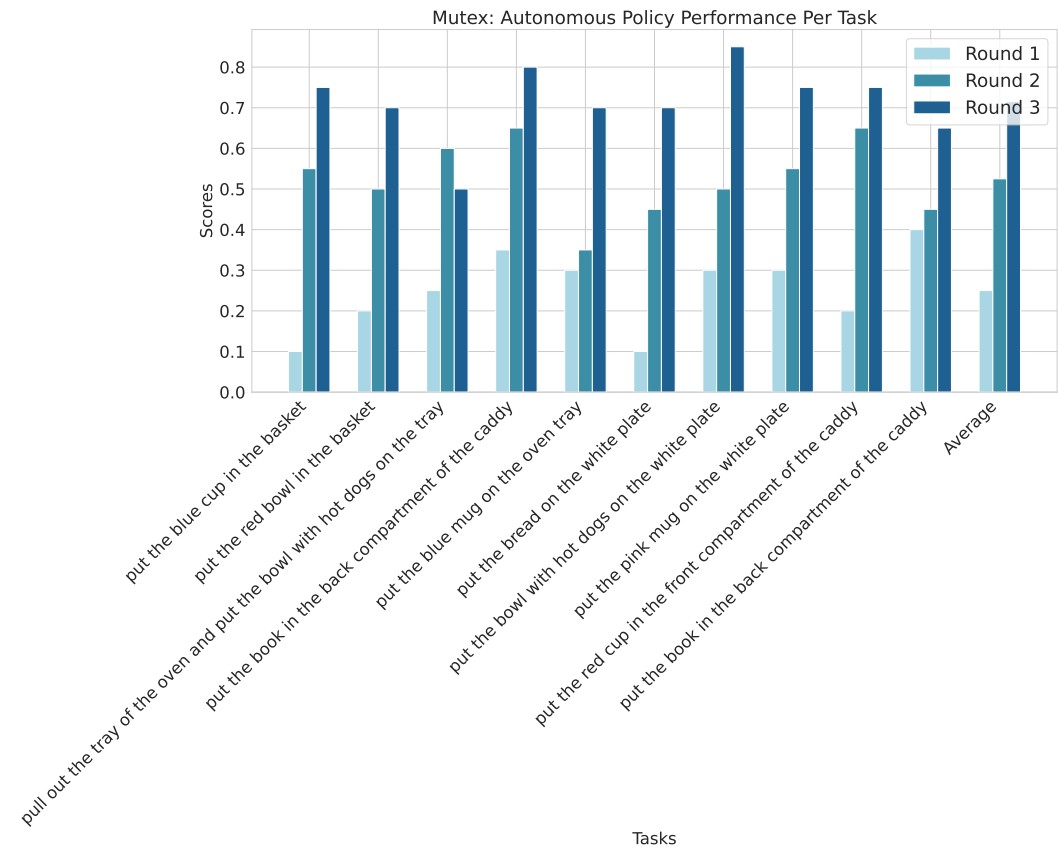

Figure 17: **Mutex: Autonomous Policy Performance Per Task.** The autonomous policy success rate improves over three rounds of deployment for the Mutex tasks.

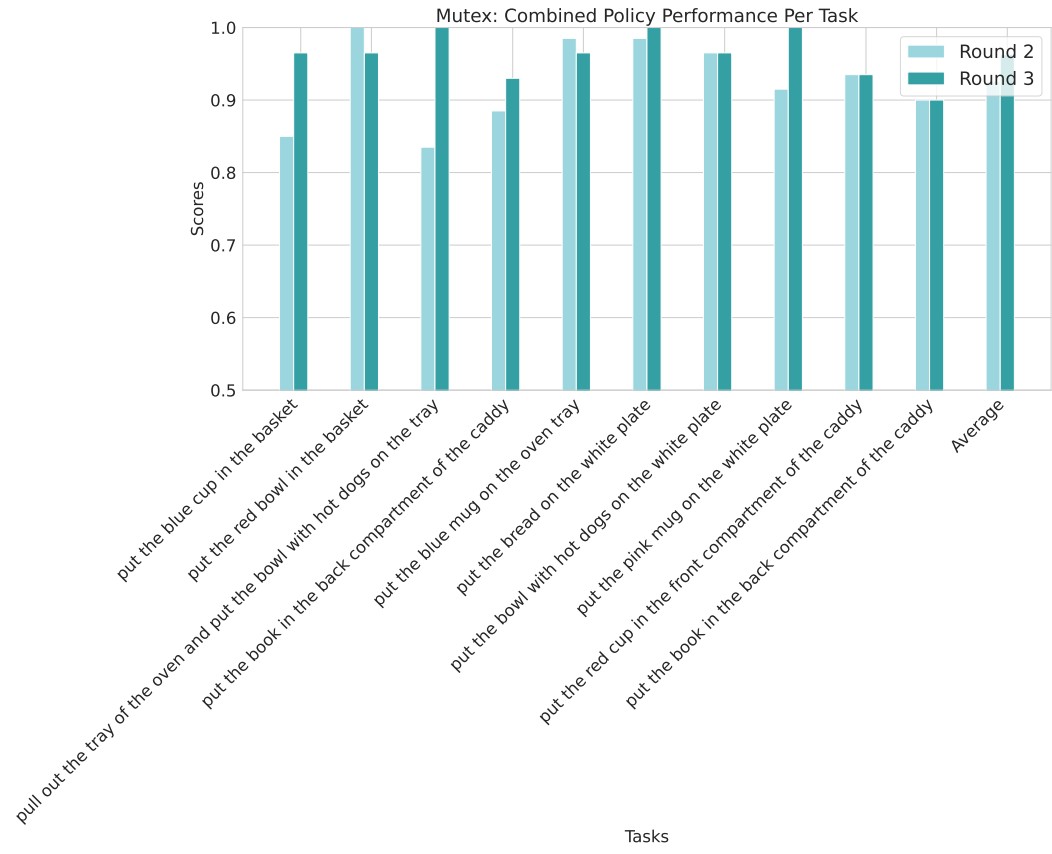

Figure 18: **Mutex: Combined Policy Performance Per Task.** The combined policy success rate improves over three rounds of deployment for the Mutex tasks.

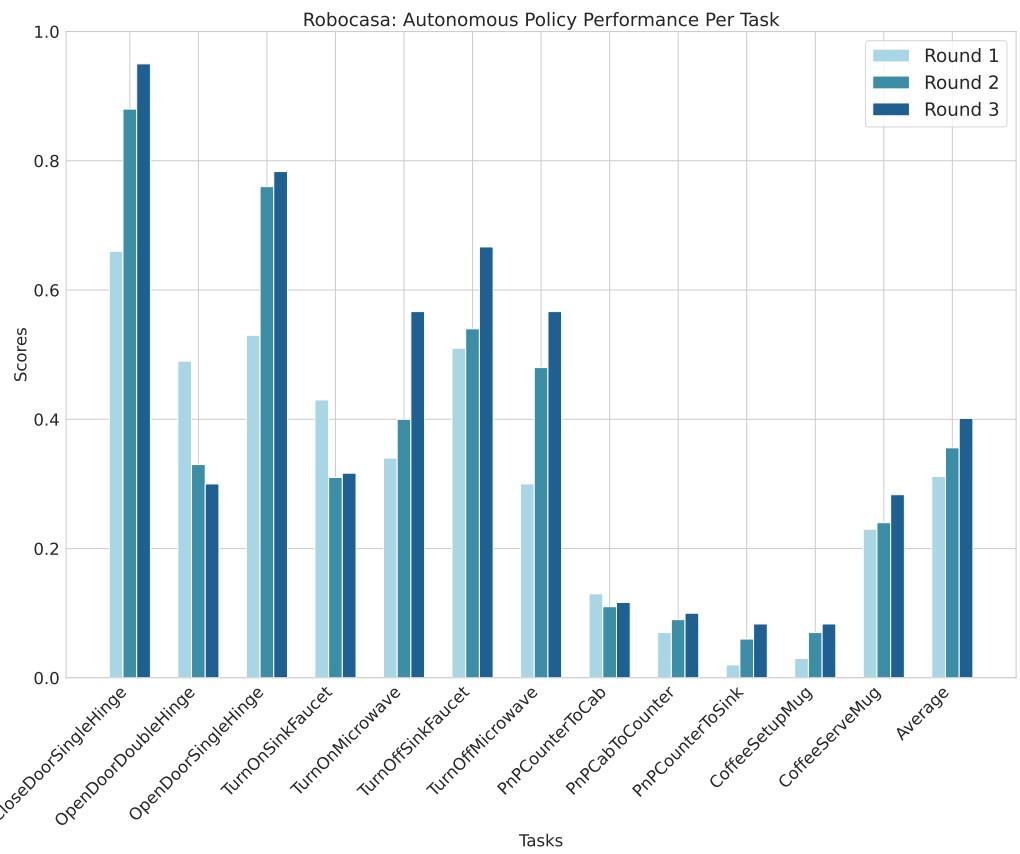

Figure 19: **RoboCasa: Autonomous Policy Performance Per Task.** The autonomous policy success rate improves over three rounds of deployment for most of the RoboCasa tasks.

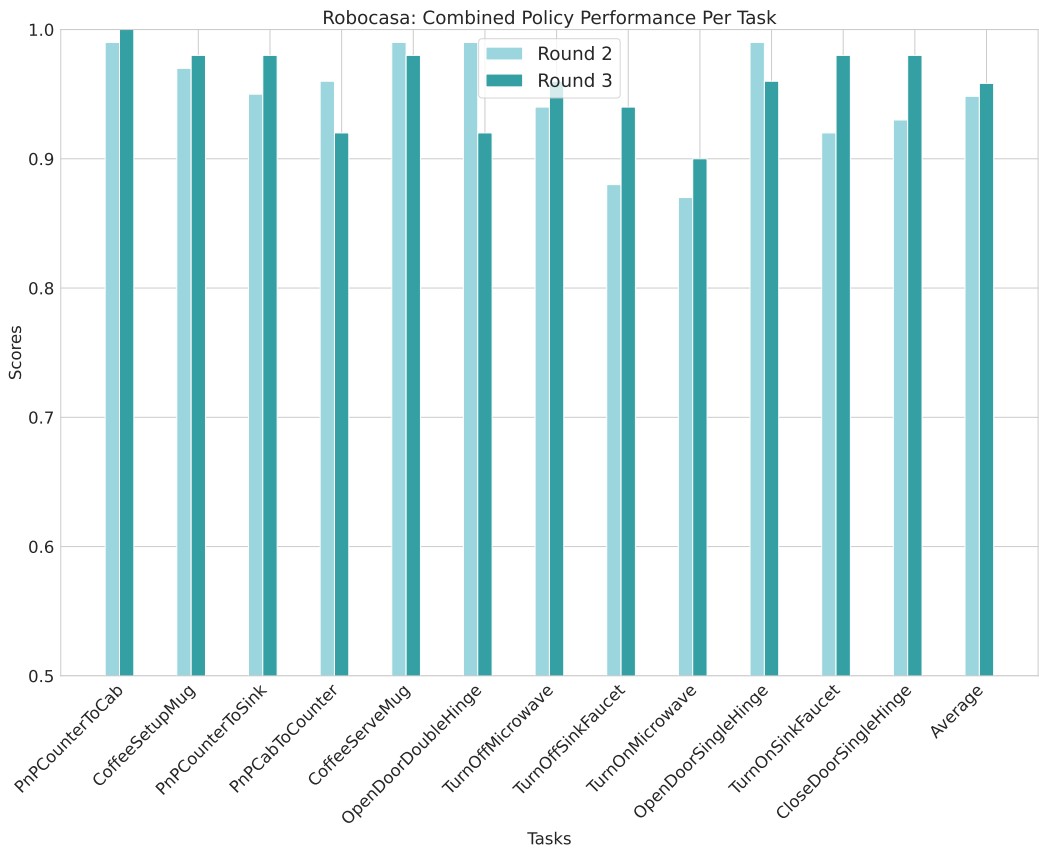

Figure 20: **RoboCasa: Combined Policy Performance Per Task.** The combined policy success rate improves over three rounds of deployment for most of the RoboCasa tasks.

