# OpenReview forum: "Multi-Task Interactive Robot Fleet Learning with Visual World Models"
_robot-learning.org/CoRL/2024/Conference — CoRL 2024_

### Official Review · Reviewer_eiin · 2024-07-18
**Firework Review**

**Originality:** 3
**Technical Quality:** 3
**Clarity Of Presentation:** 3
**Potential Impact:** 2
**Recommendation:** 3
**Confidence:** 4

**Review:**

Quality:

This work presents an iterative approach to training and managing an interactive robot fleet. It is well motivated and shows improvements over SOTA baselines.

Clarity:

For Figure 6. Top figure for Turn Off Microwave, what is the y axis referring to?  This figure could use some additional context to describe what conclusions a reader should draw from it. In Sec 5 Analysis and Discussion end with a sentence “We present more model analysis and discussion in the appendix. ” It was unclear to me what section of the appendix this is referring to.

Fig 6 Bottom Figure is a nice illustration, it would be helpful to have a similar set of overlays for the other baseline methods as well (MoMaRT, PATO, ThriftyDAgger), as well as snapshots of camera views at the moments of high probability failure to help the reader understand what is happening. This figure could be greatly expanded on (potentially in Appendix) to help the reader contrast your method to the other baselines.

Originality:

This work’s strength is that it shows improvements over SOTA baselines, and is a nice implementation of individual framework components, not that the problem or approach is highly novel.


Significance:

This work has shown to be SOTA for policy learning with human interventions for an interactive fleet. Hopefully the code can be released so that others can benchmark against it.

**Quality Of The Limitations Section:**

3

**Questions For Rebuttal:**

Figure 6 is not clear as written and is a missed opportunity to highlight how your method compares to the baselines.

**Robotics Focus:**

4

**Summary Of Paper:**

Firework is a framework for multi-task interactive robot fleet learning with a human in the loop and runtime monitoring. The approach uses a visual world model to predict task outcomes and is leveraged as an error predictor.  This approach is compared to MoMaRT, PATO & ThriftyDAgger, and shows improvements in terms of Combined Policy Performance and Return of Human Effort (ROHE).

**Summary Of Recommendation:**

This work has shown to be SOTA for policy learning with human interventions for an interactive fleet. The work could be made stronger by more clearly showing when this approach requires human interventions compared to baselines.

---

### Official Review · Reviewer_9gCY · 2024-07-21
**Good motivation, lacks ablations and insights on what aspect of the model works.**

**Originality:** 3
**Technical Quality:** 3
**Clarity Of Presentation:** 2
**Potential Impact:** 4
**Recommendation:** 3
**Confidence:** 4

**Review:**

**Strengths**

1. An important work with good motivation -- multitask policy learning is already of immense interest to the robot learning community and learning with human interventions (and that too minimal interventions) is a topic of close interest to the robot learning audience.

**Weaknesses**

1. Claims: The authors claim that their work proposed "A runtime monitoring mechanism that is based on visual world model backbone and comes 67 with task-adaptive error prediction thresholds;" -- which even [1] does.

2. On similar lines to comment 1 -- Is it an accurate statement to say that this work is an extension of [1] to multi-task policies? The related work section needs to have a dedicated comparison with this as the core components -- dynamics model, OOD detection, and failure detection are the same. The claim on L141-144 appears to be the same as the Runtime monitoring system in [1].


3. I would like to see what aspect of the proposed model is the bottleneck for policy training. It is unclear from the combined results if an accurate world model or accurate error prediction or accurate OOD detection is crucial. Ablations on that end should be included to provide insights so as to what component of the model is crucial (or are they all equally crucial). Refer to [1] for some idea on ablations.


4. Time analysis: What is the wall-clock time taken for FIREWORK compared to the baselines (ThriftyDagger, PATO, and MoMaRT)?


5. Similar to Fig. 12 (in appendix), I would like to see the # human interventions and # failures plot for rounds 2 and 3. This is primarily to see if these quantities decrease/increase from rounds 2 --> 3.

6. There are hardly any implementation details mentioned either in the main paper or in the appendix. The model is highly non-trivial with different components -- hence without the implementation details or pseudocode it will be incredibly difficult to reproduce this work.

7. [Overall writing of paper]: The writing of paper needs to improve. Specifically basing the narrative on top of [1] by introducing [1] as a preliminary could be one option. I found it relatively hard to read the paper (generally speaking). A lot of things were much clearer to me after going through [1]. I leave it upto the authors to revisit the writing.

8. [Very Minor concern] The real-world experiments have been pushed to the appendix. I'd highly recommend authors to move it to the main manuscript.

---
**References**

[1] Model-based runtime monitoring with interactive imitation learning, Huihan Liu et al., ICRA 2024.


-----

----

**POST REBUTTAL UPDATE**

I thank authors for their rebuttal experiments. Most of my concerns are addressed and hence I'm increasing the score to **Weak Accept**. I'm still not satisfied with their reply regarding the comparison between [1].

- Huihan Liu et al's [1] work is extendable to a multi-task policy and world model, specifically no assumption in [1]'s framework particularly prohibits from extending to a multi-task setting.The authors need to be clear about that and say that they are in fact extending the work (no harm in extending an existing work, in my humble opinion that by no means reduces a work's novelty).

- I would not suggest authors to use MSE as a metric to evaluate world models. MSE is particularly deceiving metric as shown in [2]. The correct way to show that a world model is better than other in a robot learning context is to evaluate it under a policy. Also, I'm little suspicious regarding the experiment results in Table 1 -- was the same amount of data given to the single-task WM? It does not make sense why a multi-task WM does better when compared to a dedicated WM for that task unless there's some difference in the data provided to both the models or some other disparity.

Again, since my other concerns were addressed I've decided to bump up the score to **Weak Accept** and I hope that the authors would carefully consider the post-rebuttal feedback and incorporate them in their work.

**Additional References**

[2] Lambert, Nathan, et al. "Objective mismatch in model-based reinforcement learning." L4DC 2020.

**Quality Of The Limitations Section:**

3

**Questions For Rebuttal:**

I would like the authors to address the concerns raised in the **Weaknesses** section and perform the experiments asked there (comments 3, 4, 5). In addition to that here are 2 questions on the design choice of the model and baselines that I have.

9. The authors mention that the world model is frozen. Why was this a design choice? This inherently assumes that the trajectories on which the world model is trained are the complete set of optimal trajectories, however, that need not be true. Also, when there is a deviation in the trajectory and the human intervenes and describes the right trajectory, the dynamics model needs to be aware of this deviated-but-successful trajectory. Am I missing something here?

10. For the baselines is the same architecture of the world model used? (Authors do mention that the policy architecture is the same.)

**Robotics Focus:**

4

**Summary Of Paper:**

This work proposes a framework for multi-task interactive training of robots. The goal is to automate the failure / OOD detection to seek human help. The framework consists of two parts: (1) pre-training the multi-task world model, and (2) enabling continual policy training via detecting failures and asking for human intervention. The work introduces an automatic way of detecting failures (based on a classifier) and OOD (based on k-means). The experiments are shown on a wide variety of tasks: 12 tasks in simulation, and 10 tasks on real-world Panda robot.

**Summary Of Recommendation:**

As it stands, there are several experiments that I've asked for and have some concerns on the core claims and implementation details. Based on this I am currently voting for a weak reject of the paper, however I will give the final recommendation after look at other reviews, authors' rebuttal and the discussion during rebuttal.

---

### Official Review · Reviewer_sGwF · 2024-07-28
**Use of learned World Model to predict failure and OOD is a promising idea. Shows good performance with respect to compared methods. Lack of clarity in terms of novelty. Lack of clarity in technical details. Ablations needed in experiments.**

**Originality:** 3
**Technical Quality:** 3
**Clarity Of Presentation:** 2
**Potential Impact:** 3
**Recommendation:** 3
**Confidence:** 4

**Review:**

Strengths:

The use of a visual World Model to predict failure and to identify out-of-distribution executions is a promising idea.

The paper is clearly presented at the high-level. (But important lack of clarity exists at the level of technical details.)

The presented set of results show good performance.


Weaknesses:

*Lack of clarity in technical details*

A variety of details were not clear for me.

The paper says "We train a failure classifier F on the frozen latent space for each specific task." This implies the tasks are discrete and are known beforehand? But the "language embedding" input in Fig. 2 implies task is specified as a language input, so it is not a predefined set of discrete tasks?

If the failure classifier can be trained for each task, why not also train other components for each task? Would the World Model perform better if it was task-specific?

Is the OOD prediction also task-specific? Not mentioned?

In the evaluation, how are the three rounds selected? How different are they from each other?

Not clear how the Evaluation Metrics are calculated. For example, how is the Autonomous Performance calculated, if, over all 3 rounds, the system required human intervention? Or is the Autonomous Performance calculated independent of the 3 rounds, on some other dataset after the 3 rounds are over? Or are they only calculated over the timesteps that were autonomous? These were not clarified at all.

Who were the human controllers in the experiments? Were they the authors? How was it ensured that human controllers were not biased to favor the proposed method?

What are the actions of the policy? Joint actions? End-effector motion? The text says "motor action" which implies joint space actions. Figure 2 shows end-effector motions. Not clear.

*Lack of clarity in terms of novelty*

I am not saying the work is not novel. But the Related Work section was not clear enough in its language to convince me that the work is novel. It does not say clearly: Is this the first time a World Model is used to predict failure/OOD? Is this the first time a multi-task policy is learned through interactive imitation learning? If yes, these should be clearly stated. If no, what are these other works and how does this work relate to those?

Related to the above, the significance of the "fleet" in this work is not perfectly clear. It could have very well been a single robot performing multiple tasks one after the other. There does not seem to be any reasoning on the performance of the whole fleet in the current round when making a decision about which robot to attract the human's attention to. Or is there?

*Ablations needed in experiments*
It is necessary to evaluate the significance of the particular design decisions. What would happen if the World Model was trained on particular tasks? What would happen if the policy were trained on particular tasks?

**Quality Of The Limitations Section:**

1

**Questions For Rebuttal:**

I have listed my questions above.

**Robotics Focus:**

4

**Summary Of Paper:**

This paper proposes a human-in-the-loop interactive, multi-task, continuously improving learning system for a fleet. There are two main components. First, a visual World Model is used to predict future visual states. The latents of this World Model predictions are used to predict whether a particular robot execution may be failing or out-of-distribution. Based on this the human supervisor is prompted to attend and intervene if necessary. The data generated is used to train an imitation learning policy for the next round (i.e., continuously improving).

**Summary Of Recommendation:**

Use of learned World Model to predict failure and OOD is a promising idea. Shows good performance with respect to compared methods. Lack of clarity in terms of novelty. Lack of clarity in technical details. Ablations needed in experiments.

---

### Author Rebuttal · Authors · 2024-08-12

We thank the reviewers and the area chair for your your time and valuable insights on our work. We have revised our manuscript and attached the updated PDF below. The changes are highlighted in pink. We encourage you to check out our appendix for updated ablations, implementation details and visualizations. Thank you again for your service!

---

### Decision · Program_Chairs · 2024-09-04

**Decision:**

Accept

**Comment:**

This paper proposes a framework for multi-task interactive robot fleet learning. The reviewers find the idea and the results promising. However, the reviewers have raised several concerns including a lack of clarity on various aspects in the approach, questions about the novelty with respect to the state of the art, lack of clarity in the evaluations and experiment settings, lack of ablations, details on the implementation are missing, the writing has to be significantly improved, unclear when the proposed approach requires human intervention, among others. Each of the reviewers has listed several questions to be addressed in the rebuttal.
Post-rebuttal: Most of the reviewers' concerns have been sufficiently addressed. Please address the remaining suggestions in the final version.